# Insights into the Molecular Mechanisms of NRF2 in Kidney Injury and Diseases

**DOI:** 10.3390/ijms24076053

**Published:** 2023-03-23

**Authors:** Da-Wei Lin, Yung-Chien Hsu, Cheng-Chih Chang, Ching-Chuan Hsieh, Chun-Liang Lin

**Affiliations:** 1Department of Internal Medicine, St. Martin de Porres Hospital, Chiayi 600, Taiwan; orcaking88@gmail.com; 2Department of Nephrology, Chang Gung Memorial Hospital, Chiayi 613, Taiwan; 3Kidney and Diabetic Complications Research Team (KDCRT), Chang Gung Memorial Hospital, Chiayi 613, Taiwan; 4Department of Surgery, Chang Gung Memorial Hospital, Chiayi 613, Taiwan; m7021@cgmh.org.tw (C.-C.C.); jeffrey570404@gmail.com (C.-C.H.); 5School of Traditional Chinese Medicine, College of Medicine, Chang Gung University, Taoyuan 333, Taiwan; 6Kidney Research Center, Chang Gung Memorial Hospital, Taipei 105, Taiwan; 7Center for Shockwave Medicine and Tissue Engineering, Chang Gung Memorial Hospital, Kaohsiung 833, Taiwan

**Keywords:** nuclear factor erythroid 2-related factor 2 (NRF2), reduction-oxidation (redox), acute kidney injury (AKI), chronic kidney disease (CKD), reactive oxygen species (ROS), diabetes nephropathy

## Abstract

Redox is a constant phenomenon in organisms. From the signaling pathway transduction to the oxidative stress during the inflammation and disease process, all are related to reduction-oxidation (redox). Nuclear factor erythroid 2-related factor 2 (NRF2) is a transcription factor targeting many antioxidant genes. In non-stressed conditions, NRF2 maintains the hemostasis of redox with housekeeping work. It expresses constitutively with basal activity, maintained by Kelch-like-ECH-associated protein 1 (KEAP1)-associated ubiquitination and degradation. When encountering stress, it can be up-regulated by several mechanisms to exert its anti-oxidative ability in diseases or inflammatory processes to protect tissues and organs from further damage. From acute kidney injury to chronic kidney diseases, such as diabetic nephropathy or glomerular disease, many results of studies have suggested that, as a master of regulating redox, NRF2 is a therapeutic option. It was not until the early termination of the clinical phase 3 trial of diabetic nephropathy due to heart failure as an unexpected side effect that we renewed our understanding of NRF2. NRF2 is not just a simple antioxidant capacity but has pleiotropic activities, harmful or helpful, depending on the conditions and backgrounds.

## 1. Introduction

Chronic kidney disease (CKD) is one of the leading burdens to health care worldwide. The prevalence in adults is approximately 10%, and this number is still growing. A systemic analysis from 1990 to 2017 revealed an approximate 30% increase in prevalence. In addition, 1.2 million people died from CKD in 2017, and the global all-age mortality rate from CKD has increased by about 40% since 1990 [1]. By 2040, chronic kidney disease is estimated to rise from sixteenth in 2016 to fifth in the ranking of global years of life-lost. Most of this increase is attributed to the growing prevalence of diabetes [2]. As we know, dysregulation of immune systems plays a pivotal role in the pathogenesis of most glomerular diseases [3,4]. Immune disorders can also be found in diabetic nephropathy, which is usually categorized as a metabolic disease [5,6,7]. Immunity and inflammation are always complementary and mutually dependent. Redox not only leads to oxidative stress and damage but also plays a crucial role as a signaling molecule in this complementary relationship and tightly modulates immunity and inflammatory response [8,9].

Renal tubular epithelial cells are high-energetic and consume oxygen during tubular reabsorption, urine production, and hormone secretions. Approximately 20–25% of daily cardiac output flows through the kidneys to supply ample oxygen [10]. Renal tubular epithelial cells have abundant mitochondria, which can produce a certain amount of reactive oxygen species (ROS). Low-level ROS are essential for pro-survival signaling to maintain homeostasis, cell proliferation, and growth. However, excessive ROS can be detrimental and highly reactive to biomolecules, including genomic DNA and protein, leading to oxidative stress [11]. There are many sources of ROS in the kidneys, but the major sources of oxidative stress in the kidneys come from nicotinamide adenine dinucleotide phosphate (NADPH) oxidative and mitochondria generation [12,13,14]. NADPH oxidases locate in the plasma membrane and generate superoxide for redox signaling. Mitochondria can produce ROS through a variety of pathophysiological stimuli, including angiotensin II, tumor necrosis factor α (TNF-α), integrin ligation, hyperglycemia, oxidized low-density lipoprotein (LDL), and superoxide from NADPH oxidase. Mitochondrial ROS production can result from the disruption of Fe-S clusters, inhibitory interactions with cytochrome c oxidase, and relative changes in expression of electron transport chain components [15,16,17,18,19]. In addition to ROS, the accumulation of electrophiles can trigger oxidative stress. For example, 15-Deoxy-∆-12,14-prostaglandin J2 (15d-PGJ2) is produced by shear stress in the vessels, leading to the activation of antioxidant activity in the endothelial cells [20]. Excessive ROS and electrophiles lead to oxidative stress followed by cell apoptosis, promoting further inflammation and tissue damage, forming a vicious cycle [11]. When chronic kidney diseases develop with the accompanying inflammation, the typical result is the progression to renal fibrosis [21].

NRF2, a transcription factor, plays a central role to maintain redox by upregulating the expression of the genes coding antioxidants. NRF2 has shown protective effects in numerous in vivo and in vitro experimental models of acute kidney injury, making it a promising therapeutic target in chronic kidney disease (Table 1). Through its transcriptional targets, NRF2 activation orchestrates a comprehensive protective response that facilitates adaption and survival under diverse biological stress [22]. From integrated transcriptomic and proteomic analyses of kidney tissue from wild-type and *NRF2* knockout mice treated with the NRF2 inducer methyl-2-cyano-3,12-dioxooleano-1,9-dien-28-oate (CDDO-Me, also known as bardoxolone methyl), kidneys of *NRF2*^−/−^ mice are found to be deficient in the expression of genes/proteins that coordinate the synthesis and conjugation of glutathione, maintain cellular redox balance, control the metabolism and disposition of a wide range of xenobiotics, and regulate the supply of NADPH and other cellular fuels. These mice also exhibited decreased levels of NADPH and glutathione in their kidneys. Overall, NRF2 regulates genes that maintain homeostasis in the kidneys, highlighting its potential as a novel therapeutic target for kidney diseases [23].

Nonetheless, the advance in clinical applications of NRF2 modulators for kidney diseases is limited by concerns of adverse cardiovascular effects. This review article will cover two main sections. The first half will review the diverse positive and negative regulators of NRF2, its interaction with endoplasmic reticulum stress, and in vivo and in vitro studies of kidney diseases, including acute kidney injury, glomerular diseases, diabetic nephropathy, and chronic kidney disease. The second half will discuss the potential pitfalls of the therapeutic application of NRF2 modulators, such as off-target effects, inadequate biomarkers, and potential adverse cardiovascular effects.

## 2. Nuclear Factor Erythroid 2-Related Factor 2

NRF2 is a transcription factor expressed in all cell types and belongs to the Cap’n’Collar (CNC) subfamily of bZIP transcription factors. It regulates the cellular defense against toxic and oxidative insults through the expression of genes involved in oxidative stress response and drug detoxification [24]. The NRF2 molecule has seven functional domains. The bZip in the ECH homology Neh1 domain heterodimerizes with small musculoaponeurotic fibrosarcoma proteins (sMAF) K, G, F, and other bZIP proteins to recognize antioxidant response elements (ARE) for activation of gene transcription, whereas Neh3, Neh4, and Neh5 are transactivation domains [25,26]. The Neh2 domain contains ETGE and DLG motifs that specifically interact with the Kelch domain of KEAP1 to mediate NRF2 ubiquitination and degradation [27]. The Neh6 domain is a KEAP1-independent degron of NRF2 that is phosphorylated by glycogen synthase kinase 3 (GSK-3), facilitating NRF2 degradation [28]. The Neh7 domain interacts with retinoic X receptor alpha (RXRα), which represses NRF2 activity [29].

NRF2 is coded by *NFE2L2*, and its expression is regulated by several transcription factors. Inflammatory stimuli, such as nuclear factor kappa-light-chain-enhancer of activated B cells (NF-κB), c-Jun, and c-Fos can upregulate the transcription of NRF2, which exacerbates its anti-oxidation ability in the inflammation. However, NF-κB p65 can compete with NRF2 for the binding to the cyclic adenosine monophosphate response element binding protein (CREBP), a transcription cofactor, which can lead to the inactivation of the NRF2 pathway (Figure 1). Additionally, NF-κB can restrain NRF2 signaling activity by recruiting histone deacetylase 3 (HDAC3) to NRF2-antioxidant response element (ARE), interacting with either CREBP or small MAF protein K (MAFK), and causing local histone hypoacetylation [30] (Figure 1). KEAP1-mediated ubiquitination and degradation of IκB kinase (IκK), another substrate, and the selective recognition by KEAP1 between NRF2 and IκK is crucial for the cross-talk between NFκB and NRF2 signaling pathways [31]. The expression of NRF2/heme-oxygenase-1 (HO-1) signaling pathway in the epithelial cells can decrease the nuclear translocation of NF-κB, suggesting the negative mutual regulation [32]. In acetaminophen-induced liver injury in mice, phosphorylated c-Jun NH2 -terminal kinase (JNK) can antagonize the cytoprotective effects of NRF2 by directly interacting with the Neh1 domain of NRF2, leading to down-regulation of ARE-driven gene expression [33] (Not shown in the figure). Both subunits of activator protein 1 (AP-1) c-Jun and c-Fos can dimerize with NRF2, but c-Jun can activate NRF2-induced transcription, while c-Fos can lead to suppression [34]. The phosphatidylinositol 3- kinase (PI3K)/Akt pathway and Notch signaling pathway can also promote the NRF2 activation and associated antioxidant functions [35,36] (Not shown in figure).

NRF2 abundance within the cell is tightly regulated and controlled by four E3 ubiquitin ligase complexes-mediated ubiquitylation and proteasomal degradation. Basal levels of NRF2 are low in unstressed conditions, mainly due to KEAP1-mediated proteasomal degradation. KEAP1 is a redox-regulated adaptor for the CUL3-RBX1 (RING-box protein 1) ubiquitin ligase complex, resulting in the ubiquitination of NRF2 in the cytoplasm and degradation of the 26 S proteasome after binding to the Neh2 domain [37] (Figure 1). The constitutive degradation of NRF2 keeps the basal expression of NRF2 target genes for housekeeping functions. During oxidative stress, electrophiles or ROS react with sensor cysteines of KEAP1, including cysteine 151 (C151), C273, and C288, leading to the conformational change of KEAP1 and, in turn, dissociating NRF2 and allowing it to escape from KEAP1-mediated degradation [38] (Figure 1). Besides the cysteine oxidation associated with a canonical KEAP1-NRF2-ARE pathway, NRF2 can be activated constitutively via another non-canonical pathway, where proteins such as sequestosome-1(p62/SQSTM1), p21, dipeptidyl peptidase III (DPP3), Wilms tumor gene on X chromosome (WTX) can disrupt the NRF2-KEAP1 complex directly [39] (Figure 1). In situations where macroautophagy is compromised, for example, p62/SQSTM1 is allowed to sequestrate KEAP1 to form an aggregate, which impairs the ubiquitination of NRF2 and leads to the constitutive activation of the NRF2 pathway (Figure 1). However, the constitutive overexpression of NRF2 may be detrimental in pathophysiologic conditions, such as chronic arsenic intoxication [40].

In addition, glycogen synthase kinase-3β (GSK3β) can phosphorylate the DSGIS motif of NRF2, which is then recognized by another E3 ubiquitin ligase, β-transducin repeats-containing proteins (β-TrCP), to facilitate the subsequent NRF2 degradation [41] (Figure 1). Additionally, GSK3β can phosphorylate Fyn, a tyrosine kinase, and make its accumulation in the nucleus. This phosphorylated tyrosine kinase, in turn, expels NRF2 out of the nucleus for degradation to prevent the accumulation of NRF2 inside the nucleus [42] (Figure 1). As a downstream unfolded protein response (UPR) product, NRF2 can be recognized by synoviolin E3 ubiquitin ligase (HRD1/SYVN1), which is expressed in parallel with X-box binding protein 1 (XBP1), an endoplasmic reticulum stress (ERS) response protein, for further ubiquitylation and degradation [43] (Figure 1). CR6-interacting Factor 1 (CRIF1), a protein previously known as cell cycle regulator and transcription cofactor, can also lead to KEAP1-independent NRF2 ubiquitylation and degradation (Figure 2). Knockout of either KEAP1 or CRIF1 can increase the level of NRF2 independently [44]. Interestingly, CRIF1 can also induce the expression of NRF2 activity via protein kinase C-δ (PKC-δ) mediated phosphorylation of NRF2 Serine 40 (Ser40) to exert the cytoprotective effects [45] (Figure 2). Another mechanism of regulating NRF2 independently of KEAP1, and repressing the cytoprotection pathway, is through RXRα inhibiting NRF2 function by directly interacting with the Neh7 domain of NRF2 in a ligand-independent manner [29,46] (Figure 2). In addition to ubiquitination and proteasomal degradation, KEAP1 can be degraded directly by chaperone-mediated autophagy (CMA), a highly selective lysosome-dependent degradation process, during prolonged oxidative stress (Figure 2). The subsequent activation of NRF2 signaling and associated antioxidative functions composed parts of the response of CMA to maintain redox hemostasis. Meanwhile, NRF2 can up-regulate the transcription of lysosome-associated membrane protein type 2A (LAMP2A), a rate-limiting factor of CMA, and form a CMA-NRF2 positive feedback loop to enhance the antioxidative response [47] (Figure 2).

After dissociation from the KEAP1 and translocation into the nucleus, NRF2 dimerizes with small MAF proteins (sMAF) or chaperones. Then it binds to antioxidant-responsive elements (ARE) in the promoters of its target genes. These target genes include NAD(P)H dehydrogenase quinone 1 (*NQO1*), heme oxygenase-1 (*HO-1*), γ-glutamyl cysteine ligase modulatory subunit (*GCLM*), the catalytic subunit (*GCLC*), and ferritin. These phase II enzymes are the primary targets of NRF2 and help maintaining a reducing environment within the cell.

## 3. Nuclear Factor Erythroid 2-Related Factor 2 and Endoplasmic Reticulum Stress

ROS are essential in regulating normal physiological functions and development at low to modest doses [48]. However, excess cellular levels of ROS can be harmful and trigger oxidative stress if they exceed the capacity of intrinsic antioxidant detoxification systems. A growing body of evidence supports the strong connection between oxidative stress and ER stress. ROS can induce ER stress. Oxidized Low-density lipoprotein (LDL) can activate ER stress to cause adverse effects in pancreatic β-cells, which is associated with type 2 diabetes through oxidative stress [49]. Furthermore, oxidative stress can cause a reduction-oxidation (redox) imbalance, worsen ER stress, as well as lead to associated apoptosis [49,50,51]. ER protein folding pathways are highly correlated with ROS production, and ER stress itself is an oxidative stress source [49,52]. Increased misfolded proteins accumulating in the ER during the pathophysiological condition lead to ER stress and activate unfolded protein response (UPR) [53]. ROS generation during UPR include oxidative protein formation with Endoplasmic reticulum oxidoreductase 1 (ERO1); protein disulfide isomerase (PDI) interaction [54], which undergoes several thiol-disulfide exchange reactions during disulfide bond modifications [55]; NADPH oxidase complex [56]; electron leakage during the transfer from NADPH to cytochrome p450 2E1 activation in ER stress and p450 endoplasmic reticulum-associated protein degradation (ERAD) [57,58]; calcium leakage from the ER; NO and high calcium influx into mitochondria with increased oxidative phosphorylation; and block the respiratory chain at complex III [59,60,61] (Figure 1).

Altered redox homeostasis in the ER can lead to ER stress, which in turn induces the production of ROS in the ER and mitochondria. To counter the deleterious impact of oxidative stress induced by protein misfolding in the ER, eukaryotic cells have evolved antioxidative stress pathways to maintain cellular redox homeostasis. During ER stress, the activation of the protein kinase R (PKR)-like ER kinase (PERK) and subsequent phosphorylation of eIF2α can lead to cell cycle arrest by inhibiting cyclin D-1 translation, along with inhibitions of 80S ribosome assembly and massive protein synthesis [62,63,64]. However, paradoxically, eukaryotic Initiation Factor 2α (eIF2α) increases the translation of activating transcription factor 4 (ATF4), which initiates a transcriptional program that includes the up-regulation of the transcription factor C/EBP homologous protein (CHOP). Downstream, ATF4/CHOP can promote proapoptotic signaling pathways. ATF4 and CHOP synergically promote the expression of genes encoding functions in protein synthesis, and these reactivated translations generate significant amounts of ROS, oxidative stress, and ATP depletion that also contribute to apoptosis [65]. Uncontrolled ER stress-mediated ROS in metabolic overload, such as diabetes, can lead to a sterile but lethal inflammatory response, resulting in renal damage in diabetic nephropathy and β cell death. Thioredoxin-interacting protein (TXNIP) is a critical factor that switches adaptive UPR to terminal UPR. TXNIP plays a crucial role in the diabetic metabolic disorder and associated ER stress by binding to NOD-, LRR-, and pyrin domain-containing protein 3 (NLRP3) inflammasomes, and activating caspase-1 cleavage, interleukin (IL)-1β secretion, and pyroptosis [66,67,68]. PERK activates TXNIP expression through downstream ATF5-associated transcription, while inositol-requiring transmembrane kinase endoribonuclease-1α (IRE1α), another branch of UPR, contributes to the regulation of TXNIP through its endoribonuclease activity that elevates TXNIP mRN. Under conditions of irremediable ER stress, IRE1α ribonuclease activity becomes less specific and triggers the decay of mi-R17, a regulator of TXNIP mRNA stability [69].

Nuclear factor erythroid 2-related factor 2 (NRF2) suppresses the basal expression of TXNIP and blocks high glucose induction of TXNIP by binding to an antioxidant response element (ARE) of the TXNIP promoter. NRF2′s binding to ARE also suppresses the binding of Mondo A, a cofactor in the induction of TXNIP, to the carbohydrate response element, with or without high glucose. As a redox regulator, NRF2 gatekeeps the basal and diabetes-induced expression of TXNIP, reducing the possibility of irreparable UPR [70]. Additionally, NRF2 is a direct substrate of PERK, where PERK-dependent phosphorylation triggers the dissociation of NRF2/KEAP1 (Kelch-like ECH-associated protein 1) complexes and translocation of NRF2 into the nucleus, up-regulating transcriptional activity. PERK activation also leads to an ATF4-dependent increase in *NRF2* mRNA, strengthening its cytoprotective function [71].

After translocating into the nucleus, NRF2 heterodimerizes with sMAF or ATF4 to regulate associated gene expression. Cells with containing a targeted deletion of ATF4 accumulate high levels of ROS, leading to cell death [72,73,74] (Figure 1), while silencing PERK impairs the import of stress-dependent NRF2 nuclear into the nucleus. Targeted deletion of *NRF2* also reduces cell survival following ER stress. Glutathione levels are dramatically decreased in *NRF2*^−/−^ cells compared with wild-type counterparts under stress or even under normal growth conditions [75,76]. NRF2′s up-regulation of phase II detoxification enzymes, such as heme oxygenase-1(HO-1), NAD(P)H quinone dehydrogenase 1(NQO1), and glutamate-cysteine ligase catalytic subunit (GCLC) contribute to the survival during the UPR response [77].

NRF2 will also suppress CHOP expression. In *NRF2*^−/−^ cells, CHOP expression is constitutively higher than in wild-type cells, and NRF2 overexpression reduces CHOP accumulation during the UPR. NRF2 acts as an intrinsic brake in ER stress to impede the spark of UPR spreading into an irremediable wildfire, leading to cell death or organ damage. However, hindering ROS production via the treatment of cells with ROS scavengers or restoration of glutathione levels delayed apoptosis, but was not competent to eliminate ER stress-mediated cell death completely [78].

## 4. Potential Therapeutic Applications in Kidney Diseases

NRF2, a transcription factor, plays a critical role in protecting against oxidative stress and regulating inflammatory response. The application of compounds that target NRF2 has piqued the interest of nephrologists. This section will review in vivo and in vitro studies conducted on various kidney diseases, such as acute ischemic reperfusion kidney injury, renal fibrosis model with unilateral ureteral obstruction, glomerulonephritis, polycystic kidney disease, and diabetic nephropathy. Some of the results will be summarized in Table 1.

### 4.1. Nuclear Factor Erythroid 2-Related Factor 2 in Acute Kidney Injury

In experiments using the renal ischemia-reperfusion model, NRF2-regulated cellular defense genes were elevated in the kidneys of wild-type mice, but not *NRF2* knockout mice. As a result, renal function, histology, vascular permeability, and survival were significantly worse in the *NRF2* knockout mice. NRF2 deficiency enhances vulnerability to both ischemic and nephrotoxic acute kidney injury [79,80]. In addition, ischemia-reperfusion injury in renal epithelial cells exacerbates ER stress and mitochondria dysfunction, leading to oxidative stress and associated injury.

X-box binding protein-1 (XBP1) is a downstream chaperone in UPR, acting as an ER stress sensor by inhibiting interaction with its co-chaperone and loss of ATPase stimulation in ER stress, switching from its chaperone cycle into an ER stress sensor cycle. Mice with XBP1^+/−^ modification in the ischemia-perfusion model showed downregulation of XBP1 and its downstream HRD1, which increased expression in the NRF2/HO-1 pathway and increased animal survival. On the contrary, the downregulation of NRF2 leads to increased ROS production, impaired kidney function, and shortened animal survival.

Upregulation of XBP1 positively correlates to increased ROS and apoptosis, but negatively correlates with proliferation in tubular cells undergoing hypoxia-reoxygenation. Overexpression of HRD1, an E3 ubiquitin ligase, promotes ubiquitination of NRF2 by binding to the motif-QSLVPDI, leading to NRF2 degradation and downregulation of HO-1 [43]. Additionally, the interaction of NRF2 and hypoxia-inducible factor-1α (HIF1α) exacerbates optimal metabolic and cytoprotective response in ischemic AKI. Hypoxia-stimulated HIF1α-dependent metabolic gene expression, as well as the mRNA of HIF-1α, were found to be significantly decreased in primary murine renal tubular epithelial cells (RTECs) with NRF2 silencing.

NRF2 is bound to the HIF-1α promoter in normoxia, but its binding decreases in hypoxia-conditions. This binding was restored soon after reoxygenation to provide the constitutive expression of HIF-1α. When NRF2 expression can be maintained by inhibiting mitochondrial complex I during hypoxic condition, the expression of HIF-1α would be sustained to provide cytoprotecting function [81].

The short-term administration of the NRF2 inducer CDDO-Me (Bardoxolone/RTA 401) during the early phase of ischemia-reperfusion injury to the kidneys was found to attenuate the late phase of tubular damage. Furthermore, this induction was shown to effectively protect the human proximal tubular cell line HK-2 from oxidative stress-mediated cell death [82]. Additionally, resveratrol, a stilbene polyphenolic compound, has been found to reduce cell apoptosis by upregulating NRF2 expression, downregulating the TLR4/NF-κB signaling pathway, and decreasing caspase-3 activity and caspase cascades in a rat model of renal ischemia and H_2_O_2_ induction [83].

In a model of sepsis-associated AKI, polydatin, another activator of NRF2, was found to mitigate the elevation of serum creatinine and BUN levels by significantly increasing the expression of NRF2 and HO-1. Additionally, it remarkably inhibited TNF-α, IL-1β, and IL-6 production, myeloperoxidase activity, and Malondialdehyde content in lipopolysaccharide-related AKI [84]. Several studies also demonstrate the beneficial effects of NRF2 enhancement and associated anti-oxidative gene expressions in other acute kidney injuries, such as pigment nephropathy, nephrotoxicity caused by heavy metals, drugs, and contrast [85].

Unilateral ureteral obstruction (UUO) is a helpful model for clarifying the pathogenesis and the mechanisms involved in progressive renal fibrosis from AKI transitioning to CKD [86]. Both oxidative stress and mitochondria dysfunction are associated with cell death and fibrosis in the UUO model. In a study of UUO after a 14-day obstruction in BALB/C mice, decreased oxygen consumption and profound mitochondrial alternations with a decrease in the abundance of respiratory complexes and mitochondrial DNA were found in both obstructed and contralateral kidneys. Intense Ca^2+^-induced swelling was elicited in the mitochondria of bilateral kidneys. Progressive decreased ROS production in mitochondria of the sham group, contralateral non-obstructed kidney, and obstructed kidneys were observed consequently. The alternations in mitochondrial structure and function in both kidneys of UUO were different from the remaining kidney after uninephrectomy. Compared with the remaining kidney in uninephrectomized mice, there was still an intriguing functional difference in respiration rate, ROS production, and abundance of antioxidant enzymes in the mitochondrial level of the non-obstructed side kidney, even though both exhibited compensatory renal growth. There were increased mitochondrial glutathione peroxidase activity and cytosolic catalase activity in the non-obstructed side kidney for the compensatory response to altered ROS homeostasis. Accumulation of phosphorylated NRF2 in the nucleus, as well as the abundance of KEAP1 in the nucleus and cytoplasm, were observed opposite to the decreased expression of HO-1, implying the dysregulation of KEAP-1/NRF2/HO-1 system in the UUO model [87]. Contrary to the prolonged UUO model, observation in early neonatal unilateral UUO rats on the 7th day revealed higher inducible heat shock protein 70 (Hsp70) levels and rapid nuclear accumulation of NRF2, KEAP1 downregulation, and mRNA induction of the NRF2-targeted genes, NQO1, and glutathione S transferase α 2 (GSTA2). These antioxidant responses faded with increased oxidative marker, lipid peroxidation, and enhanced NADPH oxidase activity after prolonged obstruction on the 14th day [88].

Regarding inflammatory pathways in the UUO model, high levels of angiotensin II stimulate NADPH oxidases (NOXs) and mitochondria, inducing ROS overproduction and favoring oxidative stress are revealed. This, in turn, activates TGF-β1 and forms a positive feedback loop further to enhance the expression of NOXs and mitochondrial ROS production, leading to the activation of several signaling pathways sensitive to the redox state such as mitogen-activated protein kinases (MAPKs), NF-κB, or Akt, and the inactivation of ROS- responsive transcription factors such as NRF2, forkhead box protein O1a(FoxO1a), and FoxO3a. These altered redox signaling pathways in the UUO model maintain the milieu for fibrosis development, ultimately progressing to CKD [89].

Pathologically, upregulation of NRF2 is demonstrated in kidney biopsies of UUO-treated mice and tubulointerstitial nephritis patients. *NRF2* knockout exacerbates early tubular damage in UUO, promotes maladaptive proliferation during the transition period, and increases fibrosis in the later stages. The deletion of NRF2 is associated with more severe injuries caused by oxidative and fibrotic inflammatory cytokines [90]. In addition, oleanolic acid has a beneficial effect, significantly reducing inflammatory cell infiltration, the ratio of Bax to Bcl-2 expression, and the number of apoptotic cells on TUNEL staining by improving NRF2 nuclear transport and reducing lipid peroxidation. This decreases renal oxidative stress and fibrosis during renal obstruction in UUO mice [91]. However, a study of the UUO model with *NRF2* knockout GFP-TG mice exhibited paradoxical findings, showing that NRF2 deficiency suppresses inflammation and fibrosis in a UUO model. In this model, M1 macrophages, instead of the M2 population, infiltrate the kidney after UUO. Persistent M1 macrophage infiltration can be continuously observed on the 14th day in wild-type mice, but there is a significant decrease in *NRF2*-KO mice. NRF2-dependent NLRP3 inflammasome activation to maintain the M1 population in this model plays a crucial role and drives the progression of CKD symptoms. After UUO, a statistically significant decrease in fibrosis was observed in NRF2 deficient kidneys, along with a decreased expression of inflammation-related genes NLRP3 and IL-1β [92]. These contradictory findings suggest there are still many unclarified variants in the UUO model, and gene expression associated with NRF2 activation is not a one-way street, varying widely across different pathological conditions.

### 4.2. Nuclear Factor Erythroid 2-Related Factor 2 in Glomerular Diseases

Imai rats, a model of spontaneous focal glomerulosclerosis, exhibit heavy proteinuria, hypoalbuminemia, hypertension, azotemia, glomerulosclerosis, tubulointerstitial inflammation, increased angiotensin II expressing cell population, up-regulations of AT1 receptor, AT2 receptor, NAD(P)H oxidase, and inflammatory mediators, activation of NF-κB, and reduction in NRF2 activity and its downstream gene expressions in the renal cortex. Treatment with angiotensin II receptor blockers prevents nephropathy, suppresses oxidative stress and inflammation, and restored NRF2 activation and expression of the antioxidant enzymes [93]. Antroquinonol, a purified compound from a mushroom, has been shown to attenuate proteinuria, renal dysfunction, and glomerulopathy, including epithelial hyperplasia lesions and podocyte injury, through the reduction in oxidative stress, leukocyte infiltration, and expression of fibrosis-related proteins in the kidney of a mouse focal segmental glomerulosclerosis (FSGS) model. Further analysis showed the benefit resulted from the increments in renal NRF2 and glutathione peroxidase activity, inhibition of renal NF-κB activation, and decreased levels of TGF-β1 in serum and kidney tissue [94]. Antroquinonol also substantially decelerated the development of severe renal lesions, such as intense glomerular proliferation, crescents, sclerosis, and periglomerular interstitial inflammation, in mice with induced accelerated and progressive IgAN (AcP-IgAN) by daily injection of purified IgA anti-phosphorylcholine antibodies and pneumococcal C-polysaccharide antigen (PnC). In advanced analysis, Antroquinonol promoted the NRF2 antioxidant pathway and inhibited the activation of T cells and NLRP3 inflammasome. Inhibition of IgA immune complex (IC) induced NLRP3 inflammasome in vitro of IgA-IC-primed macrophages by Antroquinonol was demonstrated to involve reducing ROS production [95]. In an animal model of passive Heymann nephritis (PHN) in rats, administration of curcumin significantly reduced the total cholesterol (TC), triglycerides (TG), creatinine (Scr), blood urea nitrogen (BUN), urine volume, and urine albumin. Downregulated the expression of Bax, Caspase-3, p62/SQSTM1, PI3K, p-AKT, and p-mTOR proteins and upregulated the Bcl-2, beclin1, LC3, NRF2, and HO-1 levels, as well as attenuated renal histomorphological changes were observed. The results suggest that curcumin can significantly alleviate the development of membranous nephropathy by inducing autophagy and alleviating renal oxidative stress through the regulation of PI3K/AKT/mTOR and NRF2/HO-1 pathways [96].

### 4.3. Nuclear Factor Erythroid 2-Related Factor 2 in Polycystic Kidney Disease

Tolvaptan, a vasopressin type 2 receptor antagonist, has been approved to treat autosomal dominant polycystic kidney disease. Tolvaptan was observed to activate the NRF2/HO-1 antioxidant pathway through PERK phosphorylation in renal cortical collecting duct (mpkCCD) cells and the outer medulla of mouse kidneys [97]. In an orthologous autosomal dominant polycystic kidney disease (ADPKD) mouse model, knockout of NRF2 further increased ROS generation and promoted cyst growth, while pharmacological induction of NRF2 reduced ROS production, slowed cystogenesis, and slowed disease progression [98].

### 4.4. Nuclear Factor Erythroid 2-Related Factor 2 in Chronic Kidney Disease

In 5/6 nephrectomized rats, progression of renal disease is associated with activation of the intrarenal angiotensin system, up-regulation of the oxidative, inflammatory, and fibrogenic pathways, and impaired activity of NRF2. In Sprague-Dawley rats with 5/6 nephrectomy for 6–12 weeks, pro-inflammatory and oxidative stress responses, such as lipid peroxidation, glutathione depletion, NF-κB activation, monocyte infiltration, and upregulation of monocyte chemoattractant protein-1, NADPH oxidase, cyclo-oxygenase-2, and 12-lipoxygenase, were observed in the remnant kidneys. Nuclear NRF2 activity and the associated targeted genes coding for antioxidants were reduced continuously through the 6 to 12 weeks, with a significant diminishment at 12 weeks, whereas KEAP1 was up-regulated [99].

Indoxyl sulfide, a uremic toxin, can accelerate the progression of CKD and associated cardiovascular comorbidities. In HK-2 cells treated with indoxyl sulfide, downregulation of NRF2 expression and its downstream target genes was revealed, which can be ameliorated by an inhibitor of NF-κB (pyrrolidine dithiocarbamate) and small interfering RNA specific to NF-κB p65. In vivo study with Dahl salt-resistant rats showed that indoxyl sulfide, as well as hypertension, can decrease NRF2 expression and its targeted genes, such as HO-1 and NQO1, and increase renal expression of 8-hydroxydeoxyguanosine (8-OHdG), a marker of reactive oxygen species (ROS). Administration with AST-120 can counteract the effects of indoxyl sulfide and hypertension by increasing the expression of NRF2 and its targeted gene and, meanwhile, suppressing the renal expression of 8-OHdG [100].

Increased fibrosis-related gene expression, extracellular matrix imbalance, and oxidative stress were disclosed in HK-2 cells treated with protein-bound uremic toxins, hippuric acid (HA). With NRF2 activation and knockdown, HA was approved to disrupt antioxidant networks by decreasing the levels of NRF2 with up-regulated ubiquitination and degradation, leading to ROS accumulation. ROS-mediated TGFβ/SMAD signaling, in turn, promoted the fibrotic response. The pro-inflammatory response and oxidative stress attributed to HA were further confirmed in rats, whereas treatment with sulforaphane (NRF2 activator) can reverse the HA-promoted renal fibrosis [101].

Other antioxidants, such as curcumin and synthetic triterpenoid RTA dh404, applied upon 5/6 nephrectomized mice all showed restoration of NRF2 activity and associated targeted gene expression with concomitant amelioration of inflammation, oxidative stress, and fibrosis [102,103]. In addition, treatment with cinnam-tannin A2 in a 5/6 nephrectomized rat model can further ameliorate the expression of neutrophil gelatinase-associated lipocalin (NGAL) and kidney injury molecule-1 (KIM-1), along with decreased oxidative stress by regulating NRF2-KEAP1 pathway. The tubular injury score also declined in the histopathological study of the remnant kidneys [104].

### 4.5. Nuclear Factor Erythroid 2-Related Factor 2 in Diabetic Nephropathy

Results from a biochemical analysis for 180 new-onset type 2 diabetes mellitus showed that gene expression of oxidative stress markers, such as transient receptor potential cation channel 6 (TRPC6), p22^phox^, suppressor of cytokine signaling 3 (SOCS3), and malondialdehyde (MDA) level were all increased significantly, along with T helper type 1(Th1)/T helper type 2 (Th2) ratio. In contrast, plasma NRF2 level was significantly lower than the control group [105]. Furthermore, there was a significant increase in the expression of HO-1, the ratio of Bax/Bcl-2 protein, and active caspase-3 fragments in isolated glomeruli of streptozotocin-treated rat and high glucose-treated podocytes. These increases could be inhibited by treating rats with zinc protoporphyrin in vivo or HO-1 siRNA to podocyte in vitro. The increased apoptotic cells suggest that HO-1 expression protects against podocyte apoptosis in diabetic conditions [106].

Compound mutant mice with diabetic background (heterozygous Akita strain) and knockout of antioxidant defense (*NRF2*^−/−^) showed that NRF2 deficiency could worsen diabetic kidney disease. The immunostaining and mRNA content of NQO1 were both decreased in the kidney sections of *NRF2*^−/−^ mice but paradoxically increased in diabetic background mice. The mRNA of *GCLC* (encoding glutamate-cysteine ligase) and *GSR* (glutathione reductase) was significantly low in diabetic background mice with *NRF2* knockout, and negatively correlated to increased 8-OHdG. The diabetic background mice with NRF2 knockout displayed severe diabetic symptoms, including hyperglycemia, hypertension, and albuminuria. Furthermore, uremic toxin trimethylamine N-oxide (TMAO) and an oxidative stress marker methionine sulfoxide were increased in this genotype of mice. NRF2 deficiency induced severe inflammation with macrophage infiltration in sacrificed kidneys, along with the expression of mesangiolysis and dilated capillary loops in diabetic background mice.

Finally, genetically induced NRF2 (*KEAP1^FA/FA^*) in diabetic background elderly mice showed suppression of hyperglycemia and improved tubular injury. These findings suggest that NRF2 deficiency can worsen diabetic kidney disease in diabetic model mice [107].

An animal study has demonstrated that NRF2 is crucial in attenuating renal damage in the streptozotocin-induced diabetic nephropathy model. This is evidenced by *NRF2*^−/−^ mice having higher ROS production and suffering from more significant oxidative DNA damage and renal injury than *NRF2*^+/+^ mice. NRF2-mediated protection against diabetic nephropathy is partially achieved through the inhibition of TGF-β1 and reduction in extracellular matrix production. Furthermore, the activation or overexpression of NRF2 inhibits the promoter activity of TGF-β1 in a dose-dependent manner, whereas knockdown of NRF2 by siRNA enhanced TGF-β1 transcription and fibronectin production in human mesangial cells [108]. Various pharmacological agents, such as paenolol, sodium butyrate, vitamin D, astaxanthin, isoeucommin A, dapagliflozin, and sulforaphane, have displayed benefits in ameliorating diabetic nephropathy in animal models by directly or indirectly inducing NRF2 expression [109,110,111,112,113,114,115].

**Table 1 ijms-24-06053-t001:** Application of NRF2 modulator in vivo studies and clinical trials of kidney diseases.

	Bardoxolone Methyl (BARD-Me) and Associated Analogues	Oltipraz	Dimethyl Fumarate (DMF)	Sulforaphane (SFN)	Tert-butylhydroquinone (tBHQ)
**Ischemia-Reperfusion injury**	★ Pre-treatment could reduce renal IR injury via anti-inflammatory, antioxidant, and anti-apoptotic effects [116] ★ Amelioration of ischemic AKI and increases expression of protective genes NRF2, PPARγ, and HO-1 [117] ★ protecting kidneys from ischemia-reperfusion injury in mice [118] ★ Transcription factor NRF2 hyperactivation in early-phase renal ischemia-reperfusion injury prevents tubular damage progression [82]		★ Attenuation of renal ischemia-reperfusion injury by activation of NRF2/HO-1/NQO1 signaling pathway [119] ★ The dual reno- and neuro-protective effects against uremic encephalopathy in renal ischemia/reperfusion model [120]	★ Protection against ischemia-reperfusion through induction of the NRF2-dependent phase 2 enzyme [121] ★ Sulforaphane is better than ischemic preconditioning in renal protection by activation of NRF2 [122]	★ Pretreatment alleviates IRI in diabetic rats with activation of NRF2/HO-1 signaling pathway [123]
**Sepsis/endotoxin-associated acute kidney injury**			★ Amelioration of endotoxin-induced acute kidney injury against macrophage oxidative stress [124] ★ Attenuation of LPS induced septic acute kidney injury by suppression of NFκB p65 phosphorylation and macrophage activation [125]		
**Unilateral ureteral obstruction (UUO)**		★ Protection in the UUO-mediated renal fibrosis rat model [126]	★ Attenuation of UUO-induced renal fibrosis via NF-E2-related factor 2-mediated inhibition of TGF-β/Smad signaling [127] ★ Protection against Unilateral Ureteral Obstruction-Induced Renal Damage in Rats by Alleviating Mitochondrial and Lipid Metabolism Impairment [128]	★ Preservation of mitochondrial function and suppression of UUO-induced oxidative stress, inflammation, fibrosis, autophagy, apoptosis, and pyroptosis [129]	
**Acute toxic kidney injury**	★ Amelioration of aristolochic acid (AA)-induced acute kidney injury through NRF2 pathway [130] ★ Coordinated induction of NRF2 target genes, including NQO1, GCLC, GSTpi1/2, and 4 protects against iron nitrilotriacetate (FeNTA)-induced nephrotoxicity [131]		★ Prevention of ferroptosis to attenuate cisplatin-induced acute kidney injury [132] ★ Attenuation of Di-(2-Ethylhexyl) phthalate induced nephropathy through the NRF2/HO-1 and NF-κB Signaling Pathways [133] ★ Attenuation of Calcineurin Inhibitor-induced Nephrotoxicity [134] ★ Blunting cholestasis-induced liver and kidney injury by activation of NRF2/HO-1 signaling [135]	★ Attenuation of arsenic-induced nephrotoxicity via the PI3k/Akt/NRF2 pathway in Wistar rats [136] ★ Restoration of contrast media nephropathy through NRF2/HO-1 reactivation [137] ★Attention of c ontrast-induced nephropathy in rats via NRF2/HO-1 pathway [138] ★ Protection against cisplatin-induced nephrotoxicity [139] ★ The NRF2/HO-1 system modulates cyclosporin A-induced EMT and renal fibrosis [140] ★ Attenuation of gentamicin-induced nephrotoxicity with preservation in mitochondrial function [141] ★ Prevention of maleic acid-induced nephropathy by modulating renal hemodynamics, mitochondrial bioenergetics and oxidative stress [142]	★ Alleviating contrast-induced nephropathy in rats by activating the NRF2/SIRT3/SOD2 signaling pathyway [143] ★ Preventive effect on cisplatin-induced nephrotoxicity in rats [144]
**Diabetic nephropathy**	✪ An apparent increase in kidney function following relatively short-term treatment in patients with T2D and Stage 3b-4 CKD [145] ✪ Bardoxolone methyl did not reduce the risk of ESRD or death from cardiovascular causes (BEACON trial) [146] ✪ Bardoxolone methyl significantly increased measured GFR(TSUBAKI study) [147] ★ Analogs of bardoxolone methyl worsen diabetic nephropathy in rats with additional adverse effects [148]			★ Prevention of type 2 diabetes-induced nephropathy via AMPK-mediated activation and NRF2 antioxidation [115] ★ Attenuation of experimental diabetic nephropathy involves GSK-3β/Fyn/NRF2 signaling pathway [149] Amelioration Diabetes-Induced Renal Fibrosis through Epigenetic Up-Regulation of BMP-7 [150]	★ Attenuating podocyte injury in via NRF2/HO-1 signaling pathway [151] ★ Attenuation of glomerular injury in diabetic mice [152]
**CKD induced by 5/6 nephrectomy**	★ CDDO-dhTFEA restores endothelial function impaired by reduced NRF2 activity in chronic kidney disease [103] ★ Dose-dependent deleterious and salutary actions of the NRF2 inducer dh404 in chronic kidney disease [153] ★ Restoration of NRF2 activity and attenuates oxidative stress, inflammation, and fibrosis in rats with chronic kidney disease [154]				
**Ageing associated renal injury**				★ Amelioration of age-related mitochondrial impairment and renal injury via activation of NRF2 [155]	
**Glomerulonephritis**	★ Bardoxolone methyl analog attenuates proteinuria-induced tubular damage by modulating mitochondrial function [156] ★ CDDO-Me may modulate renal damage in lupus via the inhibition of oxidative stress [157].			★ Suppression of lupus nephritis through inhibition of oxidative injury and the NF-κB-mediated inflammatory response by NRF2 [158]	
**Alport syndrome**	✪ Preservation in eGFR relative to placebo in adolescent and adult patients with Alport syndrome receiving standard of care [159]				
**Renal allograft dysfunction**				★ Improvement in chronic renal allograft dysfunction via NRF2/HO-1/NQO-1 signaling pathway [160] ★ Amelioration ischemia/reperfusion injury after Kidney Transplantation, most likely through anti-oxidative effects [161].	

✪ means clinical trials, and ★ represents in vivo studies; in vitro studies, studies of nature products, such as curcumin, and antoquinolol which is beneficial for IgA nephropathy, are not included in this table; BMP-7—Bone morphogenic protein-7; CKD—chronic kidney disease; EMT—epithelial-mesenchymal transition; GCLC—Glutamate-Cysteine Ligase Catalytic Subunit; GFR—Glomerular Filtration Rate; GSK-3β—Glycogen Synthase Kinase 3β; GSTpi—Glutathione S-transferase pi; HO-1—heme oxygenase-1; LPS—Lipopolysaccharide; NF-κB—Nuclear factor kappa-light-chain-enhancer of activated B cells; NRF2—Nuclear factor erythroid 2-related factor 2; NQO1—NAD(P)H quinone oxidoreductase 1; PPARγ—Peroxisome proliferator-activated receptor γ; SIRT3—sirtuin 3; T2D—type 2 diabetes mellitus.

## 5. Challenges of Clinical Application of NRF2 Modulators

### 5.1. Defeat from the BEACON Trial

Based on the success of many NRF2 modulators in animal models of kidney disease, clinical trials on diabetic patients have also yielded promising results in the early phase. Bardoxolone methyl (CDDO-Me), an oral antioxidant and inflammation modulator that promotes the expression of NRF2 and suppresses NF-κB, has shown a significant increase in the mean estimated glomerular filtration rate (eGFR) in patients with stage 3–4 CKD and type 2 diabetes mellitus compared to a placebo at 24 weeks [162]. However, it is important to remain cautious as the Bardoxolone Methyl Evaluation in Patients With Chronic Kidney Disease and Type 2 Diabetes (BEACON) trial, a phase 3 placebo-controlled, randomized, double-blind, parallel-group, international, and multicenter trial, enrolled 2185 patients with type 2 diabetes mellitus and stage 4 chronic kidney disease. Unfortunately, this study was terminated early due to safety concerns, with a significant increase in early heart failure events observed in patients randomized to bardoxolone methyl [146]. Additionally, the post hoc analysis revealed that an increased albuminuria was parallel with an increased eGFR [163].

Would these adverse effects be a class effect or just owing to species differences? Some animal studies or clinical trials were conducted to try to explain or make up for the imperfect results of the BEACON study. Treatment with dh404 in an Institute of Cancer Research-derived glomerulonephritis (ICGN) mouse model of nephrosis showed an apparent suppression of tubular epithelial cell damage in the renal interstitium. Dh404 treatment also maintained the number and ultrastructure of mitochondria in the tubular epithelial cells and attenuated mitochondria ROS production while these proximal tubular cells were stimulated with albumin or free fatty acid [156]. In rats with ligation of the left anterior descending coronary artery, a model of chronic heart failure (HF), short-term administration of CDDO-Me increased the stroke volume and cardiac output in CHF rats and decreased the left ventricle end-diastolic pressure found in the echocardiography. Mechanistically, CDDO-Me-induced cardiac function improvement was attributed to an increase in the expression of NRF2 and a decrease in oxidative stress in the non-infarcted areas of the heart. Additionally, CDDO-Me increased NRF2 to compete with NF-κB for binding to the cyclic adenosine monophosphate response element binding protein (CREBP), which may contribute to the selective increase in NRF2 downstream targets, including NADPH Oxidase Quinone 1, HO-1, Catalase, and GCLC Subunit, and the alleviation of myocardial inflammation in chronic HF rats. These findings suggest that NRF2 activation may provide beneficial cardiac effects in myocardial infarction-associated chronic HF [164].

Post hoc studies showed that fluid overload with heart failure occurred in the first four months after randomization in the BEACON study. For patients without baseline heart failure risks, such as baseline B-type natriuretic peptide (BNP) levels >200 pg/mL and a history of hospitalization for heart failure, the risk for heart failure among bardoxolone methyl−treated and placebo-treated patients was similar (2%) [165,166]. The Phase 2 Study of Bardoxolone Methyl in Patients with Chronic Kidney Disease and Type 2 Diabetes (TSUBAKI) aimed to determine if patients without risk factors, including baseline BNP >200 pg/mL or significant cardiovascular histories, can alleviate the fluid overload and heart failure. Inulin clearance, C_in_, was adopted to reflect actual increases in GFR, rather than eGFR. With a smaller study size enrolling only 40 patients, the TSUBAKI trial revealed that bardoxolone methyl significantly increased measured GFR, with only peripheral edema observed in four patients. Increased albuminuria, although statically insignificant, was still noticed between groups [147].

### 5.2. Theory of Endothelin Modulation

Some studies suggest that adverse cardiovascular events in patients with type 2 diabetes mellitus and stage 4 chronic kidney disease treated with bardoxolone methyl were attributed to modulation of the endothelin with the findings of meaningful reduction in urine volume and sodium excretion at week four, relative to baseline. The clinical phenotype of fluid overload and heart failure was similar to that observed with endothelin receptor antagonists in advanced CKD patients, and the preclinical data also demonstrate that bardoxolone methyl modifies endothelin signaling [165].

The conclusions of studies investigating the effect of NRF2 manipulation on endothelin expression are inconsistent. Endothelial nitric oxide (NO) synthase (NOS) in endothelial cells can be inhibited by asymmetric dimethylarginine, which is broken down by dimethylarginine dimethylaminohydrolase (DDAH). Putative antioxidant response elements for NRF2 binding are discovered in the promoters of the human *DDAH-1* and *DDAH-2*, endothelial NOS (*eNOS*), and peroxisome proliferator-activated receptor-γ (*PPARγ*) genes. Human renal glomerular endothelial cells incubated with tert-butylhydroquinone (tBHQ), an NRF2 activator, expressed increased NO, activities of NOS and DDAH, and decreased asymmetric dimethylarginine, which were exacerbated by the nuclear translocation of NRF2 to up-regulated genes of HO-1, endothelial NOS (eNOS), DDAH-1, DDAH-2, and PPAR-γ. Silencing of NRF2 diminishes these effects [167].

In an animal model of oxidative stress by slowly continuous infusion of Angiotensin II (AngII), feeding with tHBQ for two weeks alleviate the mean arterial pressure, the mesenteric arteriolar ROS, and endothelial contractility dysfunction [168]. However, another in vivo study in C57BI/6 mice treated with oral tert-butylhydroquinone (tBHQ) administration within first three days enhanced the expression of COX1, COX2, p47^phox^, and thromboxane prostanoid receptors (TPRs). Activation of TPR can increase circulating endothelin 1 (ET-1) and exacerbate the ROS and contractility responses to ET1 in microvessels [169]. Moreover, with the concomitantly continuous infusion of Ang II, these mice further exhibited the mesenteric microarteriolar mRNA for p47^phox^, the endothelin type A receptor, and thromboxane A2 synthase, as well as the increased urinary excretion of 8-isoprostane F2α. The expression of microarteriolar ROS and contractions to a thromboxane A2 (TxA2) agonist (U-46,619) and ET1 in mesenteric microarterioles were also enhanced. However, these expressions were not observed in *NRF2* knockout mice. The cyclooxygenase-1 *(COX1)* knockout genotype with the addition of blockade of cyclooxygenase-2 (COX2) and the blockade of TPRs prevented the increases in ROS and contractility. Similar findings were not observed in the follow-up of administration of tBHQ alone over 3 days in this study, or tHBQ plus Ang II slow infusion over 11–13 days in a previous study.

These findings suggest that the microarteriol contractions responding to thromboxane and ET-1 and ROS generation during an Ang II infusion in an early stage (3 days) are enhanced by tBHQ with a mechanism that depends on NRF2, COX1, and COX2 [168,170]. Therefore, NRF2 can modulate endothelin expression in the early oxidative state.

In another study aimed at treating human microvascular endothelial cells (HMEC-1), cells were incubated with 100 nM–5 μM of either bardoxolone methyl, dimethyl fumarate, or L-sulforaphane for 3 and 24 h. After a 3 h incubation, bardoxolone methyl significantly affected endothelial bioenergetics by decreasing mitochondrial membrane potential, decreasing spare respiratory capacity, and increasing proton leak. At the same time, dimethyl fumarate and L-sulforaphane did not exert such actions. However, after a 24 h incubation, bardoxolone methyl at a concentration ≥3μm decreased cellular viability and induced necrosis and apoptosis in the endothelium. Bardoxolone methyl also decreased endothelin-1 release and increased endothelial permeability even after short-term (3 h) incubation. Although all three NRF2 activators can decrease ROS production in the endothelium, bardoxolone methyl leads to a distinct endothelial profile of activity comprising detrimental effects on mitochondria and cellular viability, as well as suppression of endothelial endothelin-1 release. This interference with endothelin-1 may lead to endothelial permeability, possibly explaining the bardoxolone-associated peripheral edema [171]. However, due to the lack of endothelin signaling measurements in the BEACON study, such as urinary ET-1, the hypothesis of endothelin modification still cannot be entirely supported.

### 5.3. The More Is Not the Better

Blindly chasing the expression of NRF2 may not always lead to favorable outcomes, as even a small genetic variant can have a significant effect. For instance, NRF2 was previously thought to decelerate the progression of hepatosteatosis. When a methionine- and choline-deficient (MCD) diet for 5 days was given to *NRF2*-null, normal (wild type, WT), and NRF2-enhanced (*KEAP1*-knockdown, K1-Kd) mice, respectively. The MCD diet that caused hepatosteatosis, compared with WT mice, was more severe in the NRF2-null mice and less in the K1-kd mice. The *NRF2*-null mice had lower hepatic glutathione and exhibited more lipid peroxidation. In addition, the MCD diet activated the NRF2 signaling pathway, inducing the NRF2-targeted cytoprotective genes *NQO1* and *glutathione transferase α1/2* in WT, and even more in K1-kd mice [172]. Another study also showed that NRF2 can regulate adipocyte differentiation by controlling the expression of PPAR-γ. In vitro study with mouse embryonic fibroblasts, constitutive NRF2 activation through *KEAP1*-knockdown or treatment with sulforaphane suppressed hormone-induced differentiation and decreased PPAR-γ, CCAAT/enhancer-binding protein α, and the downstream lipogenic genes, such as fatty acid-binding protein 4 expression [173]. However, *KEAP1*-KD in C57Bl/6J, *Lep^ob/ob^* mice, an obesity animal model with leptin deficiency, even though it showed less lipid accumulation, smaller adipocytes, decreased food intake, and reduced lipogenic gene expression. Yet, enhanced NRF2 activity, impaired insulin signaling, prolonged hyperglycemia in response to glucose challenge, induced insulin resistance, and augmented hepatic steatosis were observed in an *Lep^ob^^/ob^* background. On the other hand, high fat diet-induced obesity and lipid accumulation in white adipose tissue was decreased in C57Bl/6J, *KEAP1*-KD mice without an *Lep^ob^^/ob^* background [174].

Moreover, in a study with genetically engineered partial loss of KEAP1 expression (*KEAP1^FA^^/FA^*) C57BI/6 mice undergoing unilateral nephrectomy, NRF2 was found to be constitutively activated, but baseline proteinuria was not observed unless these mice were concomitantly treated with adriamycin, angiotensin II (AngII), or a protein overload. The increased proteinuria in the *KEAP1^FA^^/FA^* mice occurs independently of GFR change. After injury, *KEAP1^FA^^/FA^* mice showed an increased glomerulosclerosis, nephrin disruption and shedding, podocyte injury, foot process effacement, and interstitial fibrosis pathologically, as well as higher daytime blood pressures and lower heart rates. Nevertheless, compound *NRF2^−/−^* mice were found to be protected from AngII-induced proteinuria. Similar to the enhancement of NRF2 in *KEAP1^FA^^/FA^* mice, treatment with CDDO-Im (a synthetic triterpenoid) at an intense dose to vigorously promote the NRF2 pathway in wild-type mice with unilateral nephrectomy did not induce proteinuria. However, CDDO-Im, in combination with AngII, greatly enhanced proteinuria dependent on the presence of NRF2 despite high glomerular expression of NQO1 and glutathione S-transferase Mu 1. Further analysis disclosed that worsening proteinuria and glomerular injury were independent of blood pressure, megalin expression, or GFR change.

Additionally, analog of the NRF2 agonist bardoxolone methyl, dh404, at a lower dose significantly lessens diabetes-associated atherosclerosis with reductions in oxidative stress and proinflammatory mediators tumor necrosis factor-α, intracellular adhesion molecule-1, vascular cell adhesion molecule-1, and monocyte chemotactic protein-1 (MCP-1) in streptozotocin induced diabetic apolipoprotein E(−/−) mice. This lower dose of dh404 also attenuates proteinuria and mesangial expansion and improves renal tubular injury. Despite the significant upregulation of the NRF2 response gene, HO-1, NQO1, and GST with the inhibition of TGF-β mediated fibrotic fibronectin, collagen I, and proinflammatory IL-6 in vitro normal rat kidney cells, a higher dose of dh404 is found to increase expression of proinflammatory mediator MCP-1 and NF-κB [175]. Similar findings are also exhibited in diabetic obese Zucker rats with 5/6 nephrectomy. Treatment with low-dose dh404 restored NRF2 activity and expression of its target genes, attenuated activation of NF-κB and fibrotic pathways, and reduced glomerulosclerosis, interstitial fibrosis, and inflammation. In contrast, treatment with a high dh404 dosage intensified proteinuria, renal dysfunction, and histological abnormalities amplified upregulation of NF-κB and fibrotic pathways, and suppressed the NRF2 system [153].

These results suggest that genetic modification for constitutive NRF2 expression or pharmacologic NRF2 activation at an intense dose may increase proteinuria in mice with CKD [176].

### 5.4. Off Target Responses

Sodium-glucose cotransporter-2 (SGLT2) inhibitor is generally accepted as a drug for renal protection. Enhanced expression of ageing marker genes (p21, p16, and p53) and senescence-associated secretory phenotypes of the kidney were observed in the *db/db* mice, compared with the *db/+* group. Treating mice with dapagliflozin but not with glimepiride can ameliorate these effects. In addition, Dapagliflozin increased the plasma level of the pre-treated β-hydroxybutyrate (β-HB), an NRF2 activator, which further promoted nuclear translocation of NRF2 and induced antioxidant pathways for anti-senescent effects [114].

In contrast, diabetic Akita *NRF2^−/−^/NRF2^RPTC (renal proximal tubular cell)^* transgenic (Tg) mice over-expressing NRF2 exhibited increased SGLT2 expression, blood glucose, glomerular filtration rate, albuminuria production, and tubulointerstitial fibrosis. These effects were not observed in Akita *NRF2*-null littermates. Treating with an NRF2 activator, oltipraz, or transfection of NRF2 cDNA enhanced SGLT2 expression and *SGLT2* promoter activity in immortalized human RPTCs (HK2) cells, and these effects were inhibited by NRF2 inhibitor, trigonelline, or NRF2 siRNA. In addition, an NRF2-RE of the *SGLT2* promoter was discovered and confirmed by gel mobility shift assay and chromatin immunoprecipitation assays, suggesting the upstream and downstream relationship. Finally, kidneys from diabetic patients showed higher NRF2 and SGLT2 in the RPTCs than those from patients without diabetes. These findings suggest that an NRF2/SGLT2 pathway exacerbates blood glucose and kidney injury in diabetes [177].

The human myocardium of dilated heart disease showed a significant increase in NRF2 with a decrease in p21-activated kinase 2 (PAK2). In C57BI/6 mice with transverse aortic constriction, PAK2 overexpression was found to enhance the XBP1-HRD1 UPR axis and attenuate tunicamycin-induced cardiac apoptosis and dysfunction. Additionally, gene array data from *PAK2^−/−^* hearts under tunicamycin-induced ER stress condition revealed dysregulated genes and altered UPR involved in glutathione metabolism and the response to oxidative stress, suggesting its participation in the redox response. *PAK2^−/−^* mice, which suffered from sustained ER stress and cardiac dysfunction, were found to increase the expression of NRF2. *PAK2* deletion reduces the protective endoplasmic reticulum function with decreased expression of XBP1. The ensuing altered proteostasis may convert NRF2 to detrimental by switching from its antioxidant role to the renin-angiotensin-aldosterone system (RAAS) gene regulator [178].

A previous study found an NRF2-responsive element in the rat *angiotensinogen (Agt)* gene promoter. In vitro, high glucose conditions, hydrogen peroxide, and oltipraz can stimulate the expression of NRF2 and the ensuing *Agt* gene. Deleting NRF2-responsive elements in the rat *Agt* gene promoter can terminate the stimulatory effect of oltipraz. Overexpression catalase can also normalize systolic BP, mitigate renal injury in Akita diabetic mice, as well as inhibit *NRF2*, *Agt*, and *heme oxygenase-1 (HO-1)* gene expression in renal proximal tubular cells at the same time [179].

Antihypertrophic properties of cardiac PAK2 were found to depend on the alleviation of ER stress by regulating the IRE1-XBP1 pathway and endoplasmic reticulum-associated protein degradation (ERAD) [180]. Meanwhile, the IRE1-XBP1 pathway has been shown to regulate autophagy both through direct transcriptional mechanism and indirect signaling pathway via forkhead box protein O1 (FoxO1) and JNK, which implied the positive correlation between PAK2 and autophagy [181]. Mice with compromised autophagy and compound *NRF2* knockout had been shown to alleviate cardiac dysfunction by preventing NRF2-mediated angiotensinogen (Agt) upregulation in response to pressure overload. When myocardial autophagy function is sufficient, NRF2 activation is cardioprotective and can inhibit myocardial necrosis in the heart. On the other hand, in mice with transverse aortic arch constriction for a setting of sustained pressure overload, constitutive cardiomyocyte-specific transgenic activation of NRF2 can suppress myocardial oxidative stress, cardiac apoptosis, fibrosis, hypertrophy, and dysfunction only when the function of autophagy is preserved. This constitutive activation of NRF2 enhances autophagosome formation and autophagic flux to clear ubiquitinated protein aggregates in cardiomyocytes and suppress proteocytotoxicity [182]. Therefore, NRF2 exacerbates cardiac maladaptive remodeling and dysfunction by enhancing angiotensin II signaling in the heart when myocardial autophagy function is insufficient. The possible mechanism may be due to the autophagy impairment mediated inactivation of Jak2/Fyn signaling for NRF2 export and degradation. NRF2, in turn, increases accumulation in the nucleus and subsequently activates NRF2-driven transcription of Agt in cardiomyocytes. Compromised ubiquitin-proteasome system (UPS) p62/SQSTM1 performance due to autophagy impairment might be another possible mechanism. This may result in sustained sequestration of KEAP1 with enhanced NRF2 accumulation and nuclear translocation [183,184].

Additionally, recent research has showed persistent NRF2 activation in non-small cell lung, in cooperation with transcription factor CEBPB, generates enhancers at gene loci that are not regulated by transiently activated NRF2 under physiological conditions. Among these unique NRF2-dependent enhancers, the NOTCH3 enhancer has been identified as critical for promoting tumor-initiating activity [185]. Furthermore, activation of NOTCH3 has been found to significantly increase in various kidney diseases, including glomerular disease, renal tubulointerstitial disease, glomerular sclerosis, and renal fibrosis. The activation also mediates these kidney disease’s occurrence and development [186].

However, it is important to note that the potential interrelationships between these findings and their implications for clinical applications require further observation and evaluation to determine any unwanted results.

### 5.5. Too Much of a Good Thing: The Reductive Stress

Overexpression of antioxidant enzyme systems promotes excess reducing equivalents that can deplete ROS, driving cells to reductive stress. These excessively reducing equivalents dampen down cell growth responses, leads to alternation in the formation of disulfide bonds in proteins, reduces mitochondrial function, and decreases cellular metabolism [187]. Although NRF2 is known as a master transcription factor that leads to antioxidant and detoxifying defense to suppress oxidative stress, some studies already demonstrate the detrimental role of NRF2 in the heart.

In an animal model with early myocardial ischemia-reperfusion for 6 h, NRF2 was found to cooperate with Programmed Cell Death 4, promoting transcription initiation of C-C Motif Chemokine Ligand 3 (Ccl3) in myocardial tissues. This, in turn, recruited C-C motif chemokine receptor 1 (Ccr1)-positive macrophages to the release of pro-inflammatory factors IL-1β and IL-6, and finally led to myocardial apoptosis and inflammatory response [188]. Furthermore, another study in transgenic mice expressing a human missense mutant of α B-crystallin (hCryAB) in cardiomyocytes formed mutant protein aggregates that sequestrate KEAP1. This resulted in persistent NRF2 activation to exacerbate ceaseless antioxidant gene expression and led to reductive stress, accumulation of ubiquitinated proteins, and pathological hypertrophy [183,189]. Nevertheless, suppressing NRF2 reduces the aggregation of mutant proteins. This implies that oxidative modification of intracellular proteins is essential to adequate ubiquitination and protein degradation. Furthermore, NRF2 deficiency is associated with significant GSH depletion in vivo and in vitro, which in turn will prevent reductive stress in the transgenic mice myocardium to prevent mutant protein aggregation and associated cardiomyopathy [190].

### 5.6. Biomarkers in Development

The KEAP1-NRF2/ARE pathway has garnered significant interest not only in kidney diseases but also in various other fields, including cancer, Alzheimer’s disease (AD), Parkinson’s disease, chronic obstructive pulmonary disease (COPD), asthma, atherosclerosis, diabetes, multiple sclerosis (MS), osteoarthritis, rheumatoid arthritis, and obesity [191]. For example, Dimethyl fumarate (DMF), an NRF2 activator, is the first Food and Drug Administration (FDA)-licensed KEAP1-NRF2 protein-protein interaction inhibitor and is used as an oral first-line therapy for relapsing-remitting multiple sclerosis [192].

The development of NRF2 modulators in translational medicine is not always smooth, as seen with bardoxolone-methyl in the BEACON trial. Nevertheless, biomarkers are crucial for drug development because they can help improve the accuracy and efficiency of disease diagnosis, prognosis, safety, and efficacy of treatment, leading to better outcomes [193]. Biomarkers are usually classified into clinical biomarkers and mechanism-specific biomarkers. The mechanism-specific biomarkers reflect the molecular action of an agent on the pharmacology or pathophysiology and target activation [194]. Current biomarkers for NRF2 modulators focus on target activation, physiological responses, and pathophysiological response, such as expression and enzymatic activity of NRF2 target genes, oxidative stress, inflammatory mediators, cells and factors, metabolomics, and carcinogen metabolite [195].

For instance, NRF2 and HO-1 are both important biomarkers involved in cellular defense against oxidative stress and inflammation, and has been adopted in several diseases, including pediatric respiratory diseases [196], cardiovascular diseases [197], movement disorders [198], malignancy [199,200], Alzheimer’s diseases, and Parkinson’s disease [201]. However, there are some limitation to their use as biomarkers. First, there is heterogeneity in the expression of NRF2 and HO-1, which can vary depending on the tissue type and disease state, making the interpretation of their significance challenging [202,203]. Second, NRF2 and HO-1 are not specific to a particular disease or condition, and their expression can be influenced by factors other than oxidative stress or inflammation [204,205]. Third, age and gender can influence the sensitivity of NRF2 and HO-1 as biomarkers. For example, studies have shown that NRF2 activity decreases with age, which may affect its sensitivity as a biomarker in older adults [206]. Therefore, it is essential to consider these limitations and interpret their expression levels in the context of other clinical and laboratory findings.

## 6. Conclusions

Redox plays a central role in inflammation and signaling conduction. NRF2 acts as a transcription factor by binding to the antioxidant response element of genes that code for antioxidant, providing defense against oxidative stress. However, molecules other than antioxidants can also be transcribed via different NRF2 response elements and, under exclusive pathophysiological conditions, these molecules may be detrimental (Figure 3). Whether NRF2 is genetically or pharmacologically induced, constitutive or intense expression of NRF2 may also be unfavorable. Reductive stress may develop in the opposite direction when the expression of NRF2 becomes extreme. Therefore, oncogenicity should be a concern when the anti-apoptosis function of NRF2 is taken into consideration. In addition, the positive and negative regulatory mechanisms of the NRF2-ARE pathway are diverse, and the expression of NRF2 intertwines with other signaling pathways such as UPR and autophagy. This results in a complicated network with crosstalk between them. Therefore, the diversity of these interactions in different kidney diseases should be considered when NRF2 target therapy is undergoing as an option of treatment. Before severe adverse effects emerge, surrogates, indices, or biomarkers of other unwanted NRF2 response element-mediated expressions, as well as a gauge of over-expression of NRF2, should be developed.

## Figures and Tables

**Figure 1 ijms-24-06053-f001:**
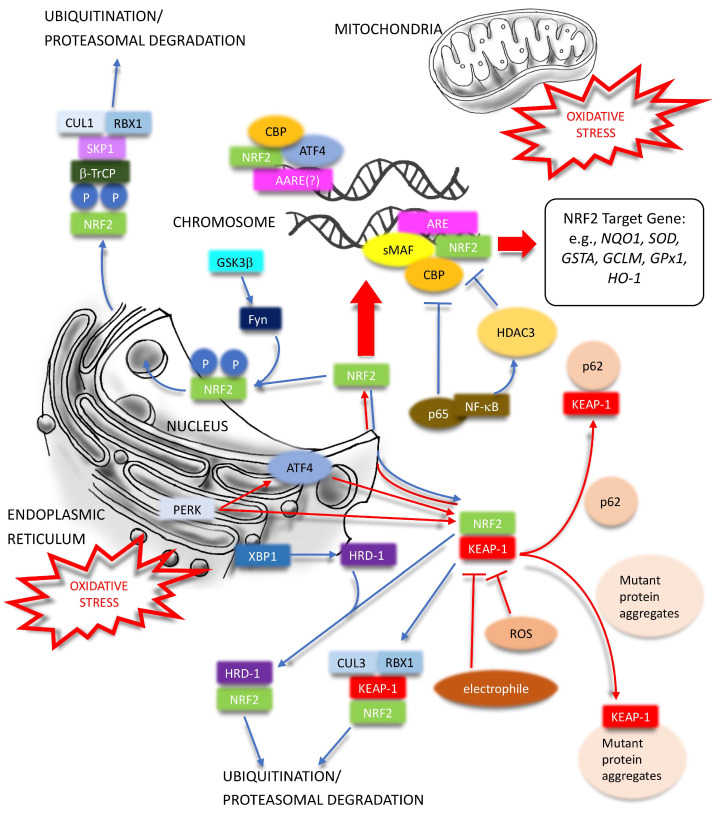
Diverse positive and negative regulation of NRF2 expression (1). Red lines denote positive and blue ones represent negative. ATF4—activating transcription factor 4; AARE—amino acid response element; ARE—antioxidant response element; CBP—CREBP/cyclic adenosine monophosphate response element binding protein; CUL—cullin; Fyn—FYN Proto-Oncogene, Src Family Tyrosine Kinase; *GSTA*—glutathione S-transferase A; *GCLM*—glutamate-cysteine ligase modifier subunit; *GPx1*—glutathione peroxidase 1; GSK3β—glycogen synthase kinase-3β; HDAC3—histone deacetylase 3; *HO-1*—heme oxygenase-1; HRD1—SYVN1/synoviolin E3 ubiquitin ligase; KEAP1—Kelch-like ECH-associated protein 1; NF-κB—nuclear factor kappa-light-chain-enhancer of activated B cells; *NQO1*—NAD(P)H quinone oxidoreductase 1; NRF2—Nuclear factor erythroid 2-related factor 2; P—phosphate; p62—SQSTM1/sequestosome-1; p65—RelA; PERK—protein kinase R (PKR)-like ER kinase; RBX1—RING-box protein 1; SKP1—S-phase kinase-associated protein 1; ROS—reactive oxygen species; sMAF—small musculoaponeurotic fibrosarcoma protein; *SOD*—superoxide dismutase; β-TrCP—β-transducin repeats-containing protein.

**Figure 2 ijms-24-06053-f002:**
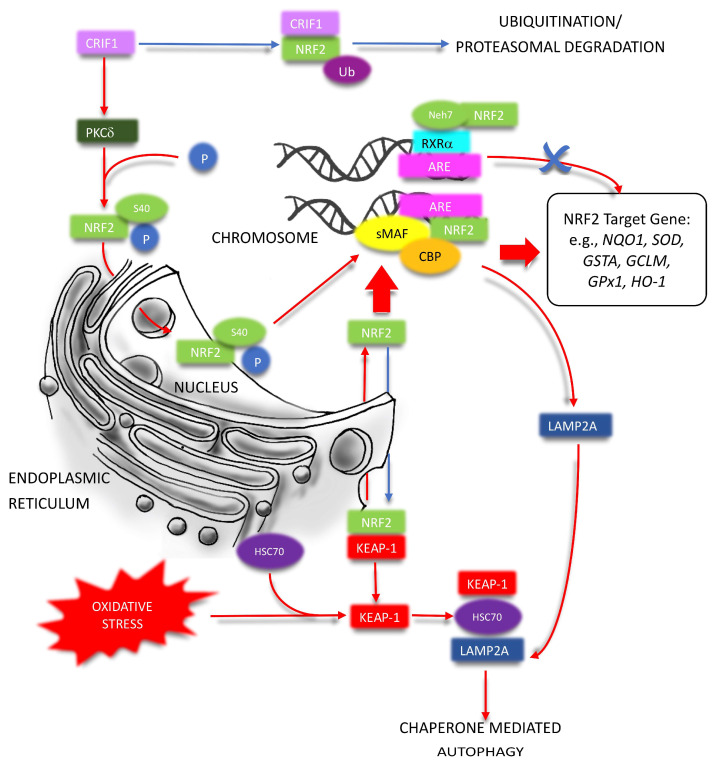
Diverse positive and negative regulation of NRF2 expression (2). Red lines denote positive and blue ones represent negative. ARE—antioxidant response element; CBP—CREBP/cyclic adenosine monophosphate response element binding protein; CRIF1—CR6-interacting Factor 1; CUL—cullin; *GSTA*—glutathione S-transferase A; *GCLM*—glutamate-cysteine ligase modifier subunit; *GPx1*—glutathione peroxidase 1; *HO-1*—heme oxygenase-1; *HRD1*—SYVN1/synoviolin E3 ubiquitin ligase; HSC70—Heat shock cognate 71 kDa protein/HSPA8; KEAP1—Kelch-like ECH-associated protein 1; Neh7— NRF2 ECH homology 7; LAMP2A—lysosome-associated membrane protein type 2A; *NQO1*—NAD(P)H quinone oxidoreductase 1; NRF2—Nuclear factor erythroid 2-related factor 2; P—phosphate; PKC-δ—protein kinase C-δ; sMAF—small musculoaponeurotic fibrosarcoma protein; *SOD*—superoxide dismutase; s40—NRF2 Serine 40; RXRα—retinoic X receptor α; Ub—ubiquitin.

**Figure 3 ijms-24-06053-f003:**
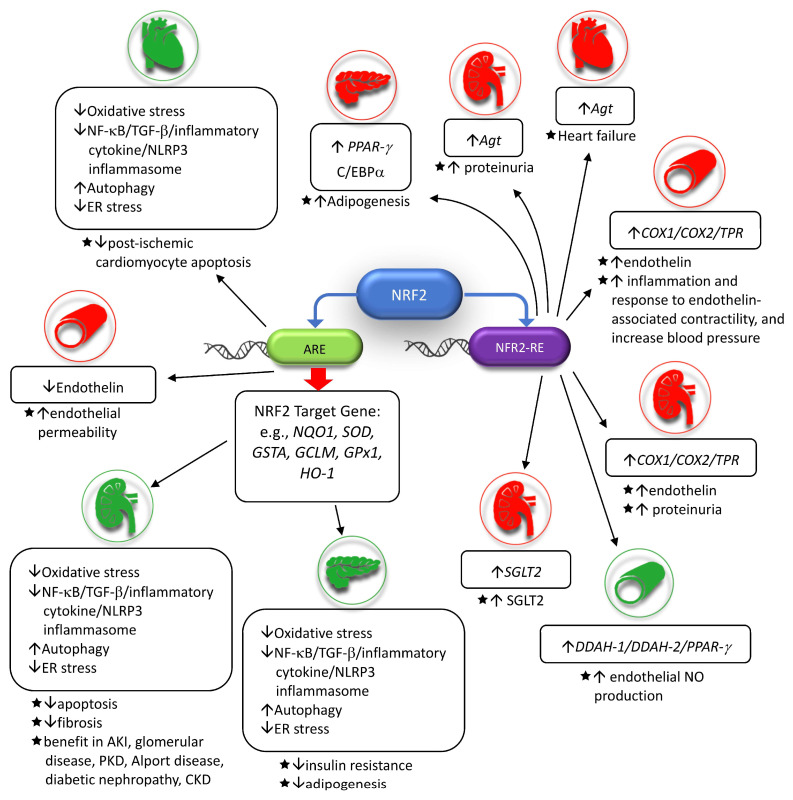
Diversity of effects via the expression of ARE and NRF2-RE. Green icons represent advantageous effects by expression of response elements (RE) mediated genes, and the red ones denote detrimental. ARE—antioxidant response element; *Agt*—angiotensin; AKI—acute kidney injury; *COX1*—cyclooxygenase-1; *COX2*—cyclooxygenase-2; C/EBPα—CCAAT-enhancer-binding proteins; *DDAH*—dimethylarginine dimethylaminohydrolase; ER—endoplasmic reticulum; *GSTA*—glutathione S-transferase A; *GCLM*—glutamate-cysteine ligase modifier subunit; *GPx1*—glutathione peroxidase 1; *HO-1*—heme oxygenase-1; NF-κB—nuclear factor kappa-light-chain-enhancer of activated B cells; NLRP3—NOD-, LRR- and pyrin domain-containing protein 3; *NQO1*—NAD(P)H quinone oxidoreductase 1; NRF2—Nuclear factor erythroid 2-related factor 2; *PPAR-γ*—peroxisome proliferator-activated receptor-γ; RE—response element; SGTL2—sodium–glucose cotransporter 2 inhibitors; *SOD*—superoxide dismutase; TGF-β—transforming growth factor-β; TPR—thromboxane prostanoid receptor.

## Data Availability

Not applicable.

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
