# Peer review of "Insights into the Molecular Mechanisms of NRF2 in Kidney Injury and Diseases"

_ijms, 2023, doi:10.3390/ijms24076053_

Round 1

Reviewer 1 Report

In the manuscript presented, Da-Wei Lin comprehensively describes the role of NRF2 in kidney injury and disease. The authors have reviewed its significance in detail as much as possible, so as an addendum that may improve the quality of the review I suggest:

1. to consider in more detail the importance of NRF2 as a target for the development of new drugs,

2. The significance of NRF2/HO-1 axis as a biomarker not only in kidney injury but also in other diseases. 

Reviewer 2 Report

 The authors have written a very up-to-date review about the molecular aspects and mechanism behind NRF2 and renal disease. The are some aspects that can be improved, in order to raise the quality of this manuscript, therefore I recommended:

Major points,

1.      The first chapter should also, briefly introduce de concept of NRF2 and its connection with renal disease; additionally, at the end of introduction the authors should insert the purpose of this review, its aims and directions.

2.      I suggest author to include a table, in which it is summarized the role of NRF3 modulators (especially the data encompassed in chapter 9) as therapeutic potential in the mentioned renal disorders.

3.      Chapter “9. Too much of a good thing” for a better overview and easy reading I recommend the authors to include subsections according to the mechanism or the targeted NRF2 pathway discussed. Example Subsection 9.1 Antioxidant and inflammation NRF2 modulators.. or  9.2 “SGLT2 and Hyperglycemia ..

Minor points,

4. There are long paragraphs without references, for example from lines 50 to 62. The sentences need to be short and concessive to the subject and have relevant references at the end.  For example, in Line 40, after “prevalence of diabetes…” there should be a reference inserted; also at line 141, after at the end of this sentence: “Oxidized Low-density lipoprotein …”.

5. Line 133-135, this phrase “This section may be divided by subheadings. It should provide a concise and precise description of the experimental results, their interpretation, as well as the experimental conclusions that can be drawn” should be deleted.

6. In Line 174, the meaning of “TXNIP” abbreviation should be inserted as it first appears in the text.

7. Line 302, the meaning for enhanced nicotinamide 302 adenine dinucleotide phosphate reduced (NADPH), was already explained in Line 152.

Line 347, does “ FSGS” stand for  Focal segmental glomerulosclerosis?; its meaning should be included.

8. Check again for any grammar and punctuation mistakes throughout the manuscript.

Round 2

Reviewer 2 Report

 It seems the authors have taken into consideration my observations, completed, and corrected the manuscript as suggested.